# Metabolomic Profiling of Obese Patients with Altered Intestinal Permeability Undergoing a Very Low-Calorie Ketogenic Diet

**DOI:** 10.3390/nu15245026

**Published:** 2023-12-07

**Authors:** Francesco Maria Calabrese, Giuseppe Celano, Giuseppe Riezzo, Benedetta D’Attoma, Antonia Ignazzi, Martina Di Chito, Annamaria Sila, Sara De Nucci, Roberta Rinaldi, Michele Linsalata, Mirco Vacca, Carmen Aurora Apa, Maria De Angelis, Gianluigi Giannelli, Giovanni De Pergola, Francesco Russo

**Affiliations:** 1Department of Soil, Plant and Food Science, University of Bari Aldo Moro, 70126 Bari, Italy; giuseppe.celano@uniba.it (G.C.); mirco.vacca@uniba.it (M.V.); carmen.apa@uniba.it (C.A.A.); maria.deangelis@uniba.it (M.D.A.); 2Functional Gastrointestinal Disorders Research Group, National Institute of Gastroenterology IRCCS “S. de Bellis”, 70013 Castellana Grotte, Italy; giuseppe.riezzo@irccsdebellis.it (G.R.); benedetta.dattoma@irccsdebellis.it (B.D.); antonia.ignazzi@irccsdebellis.it (A.I.); michele.linsalata@irccsdebellis.it (M.L.); 3Center of Nutrition for the Research and the Care of Obesity and Metabolic Diseases, National Institute of Gastroenterology IRCCS “Saverio de Bellis”, 70013 Castellana Grotte, Italy; dichitomartina@gmail.com (M.D.C.); annamaria.sila@irccsdebellis.it (A.S.); sara.denucci@irccsdebellis.it (S.D.N.); roberta.rinaldi@irccsdebellis.it (R.R.); giovanni.depergola@irccsdebellis.it (G.D.P.); 4Scientific Direction, National Institute of Gastroenterology IRCCS “S. de Bellis”, 70013 Castellana Grotte, Italy; gianluigi.giannelli@irccsdebellis.it

**Keywords:** obesity, low-grade inflammation, intestinal permeability, ketogenic diet, metabolomics, volatile organic compounds

## Abstract

A healthy intestinal permeability facilitates the selective transport of nutrients, metabolites, water, and bacterial products, involving cellular, neural, hormonal, and immune factors. An altered intestinal permeability indicates pathologic phenotypes and is associated with the exacerbation of obesity and related comorbidities. To investigate the impact of altered permeability in obese patients undergoing a calorie-restrictive dietary regimen (VLCKD), we collected urinary and fecal samples from obese patients with both normal and altered permeability (determined based on the lactulose/mannitol ratio) before and after treatment. The analysis of volatile organic compounds (VOCs) aids in understanding the metabolites produced by the intestinal microbiota in this unique ecological niche. Furthermore, we examined clinical and anthropometric variables from the cohort and compared them to significant VOC panels. Consequently, we identified specific markers in the metabolomics data that differentiated between normal and altered profiles before and after the diet. These markers indicated how the variable contribution specifically accounted for interleukins and lipopolysaccharides (LPS). The targeted metabolomics experiment detected no differences in measured short-chain fatty acids (SCFA). In summary, our study evaluated metabolomic markers capable of distinguishing low-grade inflammation conditions, exacerbated in more advanced stages of obesity with altered intestinal permeability.

## 1. Introduction

The obesity epidemic is on the rise globally, with a distribution that extends to non-westernized countries where the costs to society have not been wholly monitored yet and are related to the unavailability of a variety of healthy foods [1,2]. The energetic imbalance between consumed and expended calories is an alarming issue that creates a weight bias in healthcare systems [3].

Among non-communicable diseases (NCDs), obesity is intricately linked to a range of comorbidities that characterize its phenotypic traits, particularly concerning gastrointestinal (GI) alterations. These include changes in the integrity and function of the intestinal barrier, closely associated with long-term, low-grade inflammation, disruptions in the composition of the intestinal bacterial flora, and the production and secretion of intestinal endocrine substances—all influenced by dietary factors. These phenomena are interconnected because the intestinal bacterial flora can induce a certain degree of intestinal inflammation, leading to a disturbance in the barrier function and altered intestinal permeability. These changes, in turn, create conditions conducive to the passage of bacteria or bacterial fragments with systemic inflammatory and metabolic effects [4].

Diet, encompassing caloric intake and specific food choices, can alter the composition of intestinal bacterial flora, activating intestinal inflammation. Obesity serves as a paradigmatic example of how a metabolic disease can originate at the intestinal level, with systemic consequences characterized by damage to multiple organ tissues.

Recent scientific findings have brought attention to the ketogenic diet, primarily centered on adequate protein and essential fatty acid intake, facilitating rapid weight loss while preserving muscle mass. Currently, the very low-calorie ketogenic diet (VLCKD) stands as a valuable therapeutic option in various clinical scenarios, including obesity with comorbidities, severe obesity, hepatic steatosis, and preoperative management of bariatric surgery.

VLCKD grants efficient weight loss [5], together with a decrease in inflammation levels [6], insulin resistance [7], and metabolic disorders. This anti-inflammatory effect is also a result of ketone bodies that work by activating HCA receptor (hydroxy-carboxylic acid receptor)-2 and PPAR-γ (peroxisome proliferator-activated receptor gamma) or inhibiting nuclear factor kappa B activation [8].

While knowledge of the VLCKD effect on the intestinal barrier is still limited and contrasting, our group recently confirmed its positive impact on anthropometric indices [9]. Moreover, our research highlighted that, despite the evident benefits, VLCKD may significantly affect the barrier integrity. This diet can exacerbate the already compromised intestinal permeability and induce intestinal dysbiosis, underscoring the need for a comprehensive understanding of the diet’s effects on gut health.

The assessment of intestinal barrier integrity is carried out via non-invasive strategy, including the urinary dosing of non-absorbable sugars of different sizes. The lactulose (Lac), mannitol (Man), and sucrose (Suc) assay provides insights into intestinal permeability (IP). Lac, a disaccharide, offers information about the paracellular pathway together with tight junction (TJ) integrity, whereas the monosaccharide Man offers the possibility to detail transcellular pathway status. Bacteria in the colon break down these sugars, and their 0 to 5 h urinary fractions are utilized as markers of small intestinal permeability (s-IP); in particular, the ratio between Lac and Man is widely used in humans as a functional measurement, eliminating absorption, distribution, or excretion factors [10].

The disaccharide Suc is hydrolyzed by sucrase in the jejunum, potentially serving as an index of gastroduodenal permeability. The sugar absorption test, namely the SAT test, based on the Suc estimate is commonly employed to diagnose leaky gut syndrome, characterized by intestinal mucosal barrier dysfunction and an abnormal increase in intestinal permeability [8].

Additional tests for evaluating GI barrier function include assessing serum and fecal levels of zonulin, a protein that modulates s-IP by altering TJ interaction. Intestinal fatty acid binding protein (I-FABP) [11] and diamine oxidase (DAO) [12] are potential markers of the epithelial intestinal barrier since they are immediately released in response to altered cell membrane integrity and enter the bloodstream. These tests provide a comprehensive understanding of the integrity and function of the intestinal barrier.

Moreover, it has been shown that both altered microbiota and increased gut permeability could modify the passage of endotoxins, including lipopolysaccharides (LPS) [13]. High LPS levels promote a chronic inflammatory and immune response [12,14].

The integrity of the intestinal barrier is dependent on the tight junction that is regulated in multiple ways, including cytokines activation, tumor necrosis factor-alfa (TNFα), interferon-gamma (IFNγ) signaling kinases, and cytoskeleton proteins [15]. A cascade of protein interactions allows interleukin receptors to interact with *zonula occludens* proteins.

In our previous study [16], one of the most intriguing discoveries was that not all obese patients responded uniformly to a VLCKD regarding intestinal permeability, as measured by the Lac/Man ratio. Despite all participants having normal baseline values for the ratio (lower than 0.03), after undergoing the diet, 48% (10 out of 24) demonstrated altered intestinal permeability. This data strongly indicates variations among obese patients, suggesting that VLCKD might act as a triggering factor for the impairment of the intestinal barrier in specific individuals.

Based on these premises, we investigated the impact of VLCKD in obese patients with altered and normal intestinal permeability. Our approach relies on clinical/anthropometric data combination with GC-MS volatilome profile from fecal and urine samples to identify specific VOC makers that can be linked with obesity primary factors and comorbidities. The set of statistically significant VOCs from urine and fecal samples mainly accounted for specific aldehydes and terpenes that are indicative of a specific link with inflammation and, more importantly, with oxidative stress.

## 2. Materials and Methods

### 2.1. Population Cohort and Experimental Design

This present experimental design was conducted at the Italian Center of Nutrition for the Research and Care of Obesity and Metabolic Diseases, affiliated with the National Institute of Gastroenterology IRCCS “Saverio de Bellis” in Castellana Grotte (Ba).

The study included obese participants aged 18 to 65 years, all with a body mass index (BMI) exceeding 30 kg/m^2^. Participants underwent a comprehensive evaluation involving a medical history review, physical examination, and laboratory tests. The selected cohort comprised 25 patients without irritable bowel syndrome (IBS). Based on the lactulose/mannitol ratio, this cohort was further subdivided into two sub-clusters, consisting of 14 samples indicating normal permeability and 11 samples indicating altered permeability.

During the collection of participants’ medical history, information regarding smoking and daily alcohol consumption habits was obtained. Specifically, participants were asked to adhere to the American and European guidelines, indicating whether they consumed more than two glasses of alcohol per day (or one for females) [17]. The threshold for men was set at 30 g/day, while for women, it was 20 g/day.

Exclusion criteria encompassed hypersensitivity to meal replacement, patients suffering from cerebrovascular and cardiac diseases, respiratory insufficiency, type 1 diabetes mellitus, severe gastrointestinal diseases, chronic kidney disease, pregnancy, lactation, and psychiatric issues [5,18,19]. Additionally, we took into consideration the presence of eating disorders, liver failure, substance abuse, frail elderly patients, active/severe infections, rare diseases, disorders related to mitochondrial fatty acid oxidation, and serious mental illnesses. Moreover, 15 days before starting the diet, participants were asked to immediately stop making use of use of probiotics, vitamins, drugs, or any additional supplement.

The used protocol (Prot. n. 170/CE De Bellis) received approval from the internal Medical Ethical Committee and respected the Helsinki Declaration (1964). Participants signed a written consent before their enrolment. This present study is presented within the ClinicalTrials.gov database and corresponds to the identifier NCT05477212.

### 2.2. Study Timeline and Diet Protocol

Obese patients were recruited in the time-lapse included between April and November 2022. The Gastrointestinal Symptom Rating Scale (GSRS) questionnaire was given to patients at the first recruitment visit. Subsequently, two follow-up visits were scheduled: one before the diet starts (T0) and the second after eight weeks of diet treatment. Anthropometric measurements and fasting blood, urine, and fecal sample measured data were collected at both times. Within 2–4 days from the blood sampling, patients returned both anthropometric measurements and an assessment of their intestinal dysbiosis. This last measurement was performed again in the next three days following the diet program’s completion. Patients began their customized VLCKD only after completing the entry test. Blood withdrawal was scheduled a second time at the end of the diet program.

As already stated in our previous paper [20], we used here the first two-step protocol reported by Bruci et al. [18]. A VLCKD was administered to all participants following the two-step strategy protocol from New Penta, Cuneo, Italy. A fixed per day range was established in 20–50 g and 1–1.4 g/kg of ideal body weight for carbohydrates and proteins, respectively. Also, the intake of lipids was fixed at 15–30 g per day. Moreover, nutritionists recommend drinking at least 2 L of water per day. The esteemed per day intake was lower than 800 Kcal. In order to prevent nutritional deficits, supplements enriched in micronutrients were administered for the whole dietary treatment duration. In the first step, only meal replacements with specific quantities and types of vegetables were allowed. In contrast, the second step included a protein dish to replace one of the substituted meals.

### 2.3. Clinical/Biochemical and Anthropometric Parameters Featuring the Obese Patient Cohort

Using a calibrated scale and stadiometer, body mass index (BMI) was derived by weight and height measurements taken in fasting condition and with an empty bladder. Also, after an overnight fasting phase, blood samples were collected in the morning (from 8:00 to 9:00 a.m.). The measure of the semi-quantitative concentration of acetoacetic acid was taken directly by patients by collecting themselves the first morning’s urine before and each week until the end of diet treatment (Ketur-Test, Accu-Chec, Roche Diagnostics, Monza, Italy). Ketosis status was determined by monitoring markers in urine samples collected during the diet. Because we used a semi-quantitative method, this finding confirms that ketogenesis was present in all the patients and allowed us to define their diet as ketogenic.

### 2.4. Evaluation of Intestinal Permeability

We initiated the testing process by collecting a pre-test urine sample to evaluate the presence of endogenous sugars. Subsequently, patients ingested a solution comprising Lac (10 g), Man (5 g), and Suc (40 g) dissolved in a volume of 100 mL. Urine samples were collected up to 5 h after the administration of the solution. To preserve the samples, regardless of the final volume, a 1 mL sample of 20% (*w*/*v*) chlorohexidine was added to each collection. The total volume of excreted urine was measured and recorded. After mixing, a 2 mL sample was extracted and stored at −80 °C until analysis.

Consistent with our previous methodology [16], chromatographic analysis was employed to detect and quantify in the urine samples the levels of the three sugar probes(Lac, Man, and Suc). To inspect the Ips, the marker Lac/Man ratio was derived and used in each sample. Ratios equal to or higher than 0.030 were identified as having altered small intestinal permeability (s-IP).

### 2.5. Biochemical Assays

Biochemical assessments were conducted both upon admission and at the conclusion of the diet. Both sera and crude stool samples were promptly frozen and stored at −80 °C.

Serum and fecal zonulin levels were determined using an ELISA kit from Immunodiagnostik AG, Bensheim, Germany. Serum and stool levels below 48 ng/mL and 107 ng/mL, respectively, were considered within the range of normal zonulin levels (as stated in the manufacturer’s instructions).

Serum levels of intestinal fatty acid-binding protein (I-FABP) and diamine oxidase (DAO) were assessed utilizing ELISA kits from Thermo Fisher Scientific, Waltham, MA, USA, and Cloud-Clone Corp., Houston, TX, USA, respectively.

Circulating levels of interleukin-6 (IL-6), interleukin-8 (IL-8), interleukin-10 (IL-10), and tumor necrosis factor-alpha (TNF-α) were measured using ELISA kits from BD Biosciences, Milan, Italy.

The concentration of lipopolysaccharide (LPS) was determined using the Cloud-Clone Corp. ELISA kit from Katy, TX, USA. These ELISA-based assays provided quantitative measurements of the specified biomarkers, contributing valuable data to the assessment of various factors related to intestinal health and inflammation.

Lastly, Vitamin D status was quantitatively assessed by measuring serum 25(OH)D levels using a chemiluminescence system (DiaSorin, Stillwater, MN, USA).

### 2.6. Fecal and Urine Metabolomics

Urine and fecal samples were used to evaluate the profile of volatile organic compounds (VOCs) before and after the VLCKD diet in altered and normal obese patients. The GC-MS details relative to Clarus 680 (Perkin Elmer, Beaconsfield, UK) gas chromatography coupled to a Clarus SQ8MS (Perkin Elmer) experiment procedures are reported in our previous paper [20].

### 2.7. Quantitative Analysis of Targeted VOCs in Fecal Samples

Targeted SCFA and BCFA analyses were conducted to evaluate differences before and after diet administration in each permeability group (altered and normal). Acetic, butyric, propionic, isobutyric, and isovaleric acid (Sigma-Aldrich, St. Louis, MO, USA), plus the internal standard (IS) with a final concentration of 1 mg/L, were quantified via comparison with a standard curve. We detected each compound’s concentration via a selective ion monitoring (SIM) mode [21]. SCFA peak areas were integrated, and their absolute concentration (ppm) was derived based on the calibration curve equation, including the response value relative to IS.

### 2.8. Statistical Analysis

Because of the explanatory nature of our preliminary study, which can be considered a pilot investigation on VLCKD effects and its relationship with altered permeability, we did not compute a statistical power calculation.

Sample clustering was inspected by inferring population structure, i.e., by determining the number of groups observed with and without prior knowledge of group belonging.

We used the Discriminant Analysis of Principal Components (DAPC) (http://www.biomedcentral.com/1471-2156/11/94 (accessed on 15 September 2023)). In this multivariate STRUCTURE-like statistical approach, sample variances have been partitioned into two components and precisely a between-group and a within-group component to maximize the discrimination between groups. Specifically, the DAPC was run in the first instance without superimposing group belonging to fecal and urine sample matrix, exploiting the “find.clusters” algorithm and evaluating the Bayesian Information Criterion (BIC) curve working on model selection among a set of finite models. In the second instance, the assigned dietary/intestinal permeability belonging was set as the a priori condition for each sample group. Once the two assignments were obtained, membership probabilities from the a priori and posterior statistics were compared and graphically rendered using the “assignplot” function within the adegenet R package v2.1.1 (https://cran.r-project.org/web/packages/adegenet/index.html (accessed on 5 October 2023)). As a tangible effect of the comparison, we computed the proportion of successful reassignments. The weight of each statistical variable was assessed by computing and inspecting the loading plot, including the set of variables that mostly impacted cluster separation. These results were plotted using the R “assignplot” function, part of the “adegenet” package.

Partial least square differential analysis (PLS-DA) was run using the PLSR.Anal function implemented in R MetaboAnalystR-software (version 2.0.0). The same package allowed for obtaining the VIP score plot.

Statistically significant VOCs in pairwise group comparisons emerged by applying the nonparametric corrected Wilcoxon rank-sum test (Benjamini-Hochberg) combined with a fold change analysis.

Pearson’s correlation coefficient test, computed using the R “corr.test” function (https://search.r-project.org/CRAN/refmans/correlation/html/cor_test.html, accessed on 15 September 2023), allowed for computing statistical correlations, and results were thereby plotted by using R “corrplot” (https://cran.r-project.org/web/packages/corrplot/index.html) package, accessed on 15 September 2023. Only statistically significant correlations (*p*-values < 0.01) were used to populate the graphs.

## 3. Results

Given the monitored effect of VLCKD administration in our obese patient cohort [9], we here stratify samples based on the lactulose–mannitol absorption test. Thus, patient-assessed small intestinal permeability has been advisedly used to split the whole cohort into two subgroups, accounting for patients with normal and altered intestinal permeability. Sample adherence to each one of the permeability groups has been investigated based on urinary and fecal sample volatilome profiles (GC-MS) and clinical/anthropometric variables in 25 obese patients (11M/14F; 43.5 ±13.8 years) who underwent VLCKD administration for eight weeks. In detail, our obese patient cohort included eleven (4M/7F, 46.6 ± 11.5 years) and fourteen (7M/7F 40.5 ± 13.9 years) patients with altered and normal intestinal permeability, respectively.

For simplicity, we hereafter flagged VLCKD-treated patients with altered intestinal permeability as “pre-” (before treatment, T0) and “post-altered” after eight weeks of dietary intervention, T1) samples and, consequently, to “pre-” and “post-normal” samples as those derived from patients with normal intestinal permeability before (T0) and after (T1) the beginning of diet administration.

### 3.1. Clinical Variables Impacting Sample Clustering

Clinical and biochemical gathered non-invasive biomarkers, including anthropometric measurements, inflammation and fermentative dysbiosis, levels of urinary sucrose (Su), lactulose (La), mannitol (Ma), and circulating biomarkers (zonulin, diamine oxidase—(DAO), and intestinal fatty acid binding protein (I-FABP)) parameters were inspected by multivariate statistical analyses. The distribution of sample clouds in the PLS-DA plot shows a slight shift of the post-altered samples (Figure 1A), as also confirmed by the k-means analysis, a pre-clustering algorithm which identifies a given number of clusters, by maximizing the variation among groups, then verified by the “a posterior” DAPC (Figure 1C). The lactulose/mannitol (Lac/Man) ratio, indican, and lipopolysaccharide (LPS) are the variables that mainly contributed to marking the post-altered group in both the statistical approaches used, as evidenced by the almost concordant VIP score plot (Figure 1, panel B) and DAPC variable loading plot (Figure 1, panel D). Moreover, both analyses revealed a contribution of interleukins (IL 10 in particular), which evidenced an involvement of the immunity system in those patients with altered permeability after the VLCKD administration.

### 3.2. Group Clustering Based on Urine and Fecal Volatilome Profiles

In the first instance, fecal and urinary sample VOC profiles were inspected by discriminant analysis of principal components (DAPC). Urine and feces-derived volatile compounds revealed how urine metabolomics analysis (Figure 2) allowed a better distinguishing of clouds than that observed by exploiting fecal samples (Appendix A). More specifically, the DAPC plot, based on the urine VOC matrix per sample compound concentrations, shows a spatial shift in the cloud of patients marked by altered intestinal permeability after the treatment. More precisely, this cluster is placed separately from the others in the third quadrant of the Cartesian diagram and indicates how the statistical contribution of urine VOCs most likely relied on squishing the clouds relative to normal and altered samples before the VLCKD treatment. These two clusters are also closely placed in the post-normal patient cloud.

On the contrary, fecal metabolites in the DAPC plot (Appendix A) indicated how the two groups of post-treated patients with altered and normal permeability resolved closely.

Describing the discriminant analysis, the sample reassignment (group belonging to a priori clustering model compared with a posterior DAPC assignment) indicated only two misclassified samples (Figure 2 panel B) belonging to the pre-altered and post-normal groups, respectively. In other words, the posterior DAPC classification (percentage of membership probability assignment) was consistent with the original clusters except for two discrepancies, thus sustaining almost the totality of patient group belonging.

Looking at the variable loadings, seventeen VOCs are mostly responsible for the resulting DAPC clustering (Figure 2 panel D). Specifically, in order of statistical weight, prehnitol or durene, benzaldehyde, methyl isobutyl ketone; 2(1H)-naphthalenone, 3,4,4a,5; hexanal; octanoic acid; acetone; (1,4-Dimethylpent-2-enyl)benzene; heptanoic acid, undecane, benzaldehyde, 2,5-dimethyl-, 2,5-cyclohexadiene-1,4-dione; 2-Pentanone, 1-cyclohexene-1-carboxaldehyde; benzofuran, 4,7-dimethyl-; and 2,4-dimethylfuran and m-Cymen-8-ol were over the arbitrary threshold of 0.02.

### 3.3. Urine VOC Pairwise Group Comparison in Altered and Normal Patients before and after VLCKD Administration

When metabolomics urine profiles from obese pre-altered and post-altered VLCKD-administered patients were compared via a non-parametric statistical test (Wilcoxon rank-sum test) combined with a fold change analyses, three VOCs resulted in being significantly different in their concentrations. As reported in the volcano plot (Figure 3), hexanal and gamma-terpinene significantly decreased in post-altered patients, whereas acetone increased due to VLCKD administration.

An increase in acetone and a decrease in hexanal concentrations were also found when post-normal and pre-normal patients were compared (Figure 4). In addition to these two VOCs, a statistically significant increase in 2-methoxythiophene, benzaldehyde, 2,5-dimethyl- and disulfide dimethyl and a concomitant decrease in the concentration of benzofuran, 4,7-dimethyl-marked patients with normal intestinal permeability (Figure 4). Benzaldehyde, 2,5-dimethyl-, hexanal, and acetone also contributed as impacting variables to the DAPC clustering of the four groups.

### 3.4. Fecal VOC Pairwise Group Comparison in Altered and Normal Patients before and after VLCKD Administration

When the matrix of fecal VOCs was inspected for statistically significant variables emerging from the comparison between post- and pre-altered obese patients (Appendix A), only propanoic acid, 2-methyl-, 2-phenylmethyl ester resulted in decreased. On the contrary, five volatile compounds resulted in an increase in their concentration after the VLCKD and precisely: anethole, styrene, propane, 2-methoxy-2-methyl-; cyclohexene, 1-methyl-4-(1-methyl ethylidene) and bicyclo[2.2.1]heptan-2-ol, 1,3,3-trimethyl-.

Noteworthy, the comparison of fecal volatile profiles of VLCKD-administered patients marked by normal intestinal permeability compared with untreated ones revealed a higher number of statistically significant VOC (Appendix A). The list of six significantly decreased fecal VOCs included hexanoic acid butyl ester, pentanoic acid butyl ester, phenol, 3-(1-methylethyl)-, methylcarbamate, pentanoic acid propyl ester, cyclohexene, 1-methyl-4-(1-methylethylidene)-, pentadecane, 2,6,10,14-tetramethyl. A more conspicuous set of VOCs increased in concentration (25 in total) accounted for 1,2-propanediol; nonanoic acid; hexadecane; caryophyllene; ethylamine; trimethylamine; heptadecane; 1-undecanol; alpha-muurolene; phenol, 3,4-dimethoxy-; cyclopentadecane; menthol; tetradecane; 1-tetradecanol; naphthalene, 1,2,3,5,6,8a-hexahydro-4,7-dimethyl-1-(1-methylethyl)-; acetophenone, 2-chloro-; 1-Hexanol, 2-ethyl-; carbamic acid, methyl-, 3-methylphenyl ester; naphthalene, 1,2-dimethyl-; bicyclo[2.2.1]heptan-2-ol, 1,3,3-trimethyl-; propylene oxide; butanoic acid, 3-methyl-, ethyl ester; 2,3-dihydroxy-3-methylbutyric acid; ethanone, 1-(2-hydroxyphenyl)-; and methyl isovalerate.

### 3.5. SCFA Profiles from Targeted GC-MS

Concentrations of isovaleric, butanoic, isobutyric, propanoic, and acetic acids were detected using a GC-MS targeted experiment. No one of the measured SCFA changed in its concentrations when altered, and normal permeability sample groups were compared after and before VLCKD by means of a two-group corrected Wilcoxon sum rank test (or also in a multiple test corrected Welch test). The raw detected concentration values of SCFA are reported in Appendix A.

### 3.6. VOC Clinical/Anthropometric Variable Pearson Correlations

After obtaining the statistically significant set of urine metabolomics compounds, we focused on the link between VOCs and clinical/anthropometric variables (Appendix A). Regarding the VOC dataset to be used as a matrix in the correlation with clinical/anthropometric data, our choice relies upon the best set helpful in predicting a shift in the post-altered sample clouds in the multivariate analysis. Following this rationale, only urine VOCs take part in the correlation analysis.

Our Pearson correlation (Figure 5) shows that hexanal positively correlates with the lactulose–mannitol ratio and vitamin D (Vit D). Vitamin D, in turn, increases together with 2-methoxy thiophene levels. Moreover, disulfide dimethyl is negatively correlated with IL10 and instead was positively correlated with acetone. Among positive correlations, 2-methoxy thiophene was linked with two anthropometric variables, i.e., BMI and waist circumference.

## 4. Discussion

It is well known that host genetics, diet, and intestinal microbiota make the three-way basis of a batch of factors involved in host metabolism regulation and related changes in gut microbiota that ultimately predispose to obesity and related pathologies with overlapping phenotypes [22]. Also, recent proof of evidence revealed how high-fat, unbalanced diets varied in carbohydrate content, coupled with unique combinations of host genetics, drive changes in gut microbiota composition [23].

It is important to consider that specific VOC detection can be related to the presence of specific taxa and gut-microbiota-related metabolites. Both overweight and obese patient conditions and diet change the balance in low-molecular-weight metabolites whose production undergoes specific regulation.

In direct connection with the host’s physiology and intestinal homeostasis, the inspection of fecal and urine volatile profiles of obese patients proves handy for identifying specific metabolites derived both by the host’s microbiota and the host’s metabolism [24]. Specific VOCs may serve as markers of specific host-associated phenotypes, functional in disclosing specific progression of symptoms related to obesity status.

In this context, we examined the impact of a very low-calorie ketogenic diet (VLCKD) on a cohort of obese patients by comparing samples collected before and after an 8-week dietary intervention [20]. This dietary regimen influenced metabolic pathways, resulting in a reduction in glycolysis and an elevation in lipolysis, glycogenolysis, and gluconeogenesis [25].

Our objective in this study was to explore whether significant changes in volatile organic compound (VOC) profiles were evident in two distinct sub-cohorts of obese patients characterized by altered or normal intestinal permeability following the administration of VLCKD.

To accomplish this, we initially stratified patients based on lactulose/mannitol ratio measurements and collected urine and fecal samples, which were then analyzed via an untargeted GC/MS experiment. Concurrently, we monitored SCFA concentrations using a targeted GC/MS approach.

As for the initial results, we did not observe statistically significant changes in the levels of acetic, propanoic, isobutyric, butanoic, and isovaleric acids when comparing post-VLCKD with pre-VLCKD patients in both sub-groups, i.e., those with altered and normal intestinal permeability.

As revealed by our cluster analyses (including PLS-DA and DAPC), without compelling samples to adhere strictly to their respective groups, the set of clinical and anthropometric parameters enables the grouping of post-altered samples to be visually separated from the others.

As expected, the statistical importance of variables, as indicated by the PLS-DA VIP graph and DAPC loading plot, in the post-altered patient sample group demonstrated a more substantial contribution from factors such as the Lac/Man ratio, urinary indican levels, circulating concentrations of LPS, various interleukins, DAO, IFAB, and vitamin D. These results support and confirm the existence of (i) fermentative/putrefactive dysbiosis, (ii) chronic low-grade inflammation, and (iii) specific phenotypes associated with obesity, i.e., potential liver steatosis, a comorbidity trait of obesity, (iv) the passage of endotoxins due to increased permeability and (v) a related dysmetabolic state of visceral adiposity. Dealing with the fifth listed point, this dysmetabolic state relies on decreased Vitamin D levels, which are a biomarker of visceral adiposity [26].

A comparable spatial distribution cluster, akin to that yielded by clinical and anthropometric discriminant variable analyses, was achieved using the set of urine volatile organic compounds (VOCs). Once again, this enables segregating the post-altered samples from the others. This observation suggests a more pronounced impact of dietary treatment on patients with altered intestinal permeability, a pathological condition accompanied by a low to medium grade of inflammation detected via urine volatile compounds. The link between increased intestinal permeability and variations in the urinary concentration of specific metabolites has been explored and discussed in the literature [27] and needs to take into account the age, gender, and lifestyle of the patients.

In the context of lipid peroxidation, the chronic inflammation induced by LPS has been shown to impact hyperinsulinemia, exacerbate inflammation [28], and modify lipoprotein profiles, subsequently affecting intestinal permeability.

Classically present in blood, urine, and breath during fasting and caloric restrictions [29,30], higher acetone levels in post-altered urine metabolomics profiles indicate the sustained presence of a ketogenic metabolic state. The comparison of normalized concentrations of acetone in the altered versus normal groups highlights a generalized worsening of pathologically related conditions in obese patients with altered permeability following the VLCKD administration.

Additionally, the presence of acetone in feces has been correlated with specific gut taxa in pathologies involving altered intestinal barriers, such as inflammatory bowel disease (IBD) [31]. Conversely, two volatile organic compounds (VOCs), hexanal and gamma-terpinene, indicative of lipid peroxidation and decomposition of linoleic acid [32], exhibited decreased concentrations. Gamma-terpinene, a key monoterpene with antibacterial and antioxidant properties found in various essential oils [33], indicated an increased burden of oxidative stress in patients with altered intestinal permeability.

Consistent with other published evidence highlighting the antioxidant and anti-inflammatory effects of the ketogenic diet [34], our previous study also reported a decrease in hexanal after consuming VLCKD [20]. Notably, the panel of significant urine VOCs reported in patients with normal permeability included specific aldehydes (2-methoxy thiophene, benzaldehyde, 2,5-dimethyl-, and disulfide dimethyl) linked to lung [35] and breast cancer [36].

Patients with normal permeability also showed a decrease in the concentration of benzofuran, 4,7-dimethyl-, a compound associated with normal fluctuations in the urine of healthy individuals [37] but recognized in urine profiles from patients affected by lymphoma [38].

In the analysis of fecal volatilome, VLCKD revealed an increased concentration of anethole, styrene, propane, 2-methoxy-2-methyl-, cyclohexene, 1-methyl-4-(1-methyl ethylidene), and bicyclo[2.2.1]heptan-2-ol, 1,3,3-trimethyl-, in altered permeability samples. Anethole, for instance, has been associated with preventing the downregulation of ZO-1, occludin, and CLDN-1 in LPS-treated rat intestinal epithelial cells [39]. Similarly, a recent study reported the anti-inflammatory effect of anethole in E. coli enterotoxigenic destruction of the intestinal barrier [40], suggesting a potential role in restoring damages related to altered permeability by impacting immune system gene signaling and intestinal microbiota.

Styrene levels have been linked to an increased risk of kidney stones in the American population, particularly in individuals with overweight/obesity [41].

For the other statistically significant fecal VOCs, limited evidence is available. 1-methyl-4-(1-methyl ethylidene) is a component of *Curcuma longa* radix extract used in Chinese traditional medicine for its health-promoting effects [42]. Bicyclo[2.2.1]heptan-2-ol, 1,3,3-trimethyl-(fenchol), a terpene found in various plant oil extracts [43], has been demonstrated as a potent activator of free fatty acid receptor 2 (FFAR2) signaling, which may prevent the increase in amyloid-beta (Aβ)-stimulated neuronal toxicity in Alzheimer’s disease [44].

In contrast, the set of statistically significant fecal VOCs in normal permeability patients revealed a different effect of VLCKD. Some healthy-related esters, including hexanoic and pentanoic acid butyl esters and phenol, 3-(1-methylethyl)-, and methylcarbamate, decreased after VLCKD in normal permeability patients. Van Deuren et al. studied the effect of the administration of butyrate and hexanoate-enriched triglycerides on the postprandial systemic levels of these two esters, which tend to increase in overweight/obese patients [45]. The same authors revealed that BCFA 2-methylbutyrate, iso-butyrate, and iso-valerate did not significantly change. In our patients with normal permeability, VLCKD treatment led to a decrease in these two esters and a concomitant increase in the branched-chain SCFA methyl iso-valerate.

Furthermore, the high-ranked fold change variable set indicating a potential VLCKD health-promoting effect included butanoic acid, 3-methyl-, ethyl ester, and the conjugated base of 2,3-dihydroxy-3-methylbutanoic acid.

In our correlation analysis, a strong negative correlation between hexanal and Vit. D emerged. As demonstrated by Kong et al., increased levels of activated 1,25(OH)2D3, as found in our altered patients, lead to an increase in tight junction proteins and E-cadherin, improving healing following intestinal epithelium injury [46]. Thus, when Vit D exerts its protective effect, hexanal decreases, indicating a reduction in oxidative stress.

Our correlation analysis also indicated a negative correlation between disulfide dimethyl, a molecule with increased levels in breast cancer patients [36], and IL10, suggesting a potential link between this compound and anti-inflammatory processes.

It is worth saying that our dietary intervention study presents a first limitation dealing with the semi-quantitative method used for urine ketone body quantification. Moreover, ketone body concentration was not evaluated in blood samples. We considered our diet as ketogenic based on the ketosis level measured in urines.

A second limitation is the lack of 16S sequencing data that can be useful to trace the possible connection between VOCs and fecal microbiota taxa producers and understand the impact of this diet on the gut consortium.

## 5. Conclusions

In our present study, we explored the impact of an 8-week VLCKD on obese patients with both altered and normal intestinal permeability. Our investigation focused on identifying statistically significant volatile organic compounds (VOCs) derived from urine and fecal samples. Employing a robust statistical approach, we pinpointed VOCs likely influenced by microbiota taxa and host cells. These identified VOCs were subsequently correlated with clinically and anthropometrically measured variables.

With the markers obtained from our GC/MS approach, our future perspective involves delving into the commensal taxa of the intestinal microbiota within our obese cohort. We aim to identify key functions via a meta-transcriptomics experiment, providing deeper insights into the molecular processes underlying the observed changes. This integrated approach will contribute to a more comprehensive understanding of the complex interplay between diet, microbiota, and host physiology in the context of altered and normal intestinal permeability.

## Figures and Tables

**Figure 1 nutrients-15-05026-f001:**
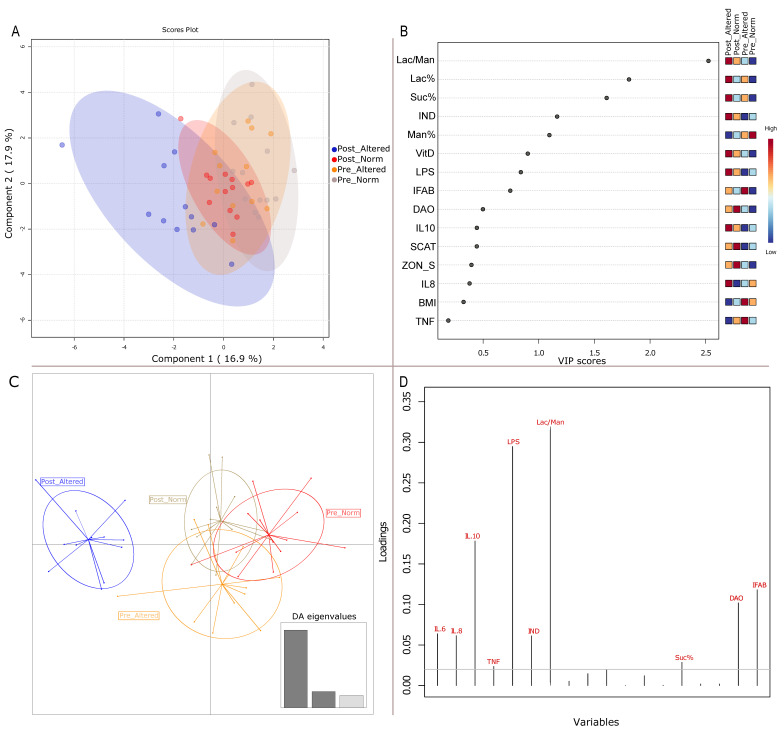
Clustering analyses on clinical/biochemical and anthropometric variables. PLS-DA and DAPC clustering based on clinical and anthropometric sample variables. (**A**) PLS-DA score plot. (**B**) VIP scores of most impacting variables. Red to blue squares indicate high and low variable contributions in each of the groups, respectively. (**C**) DAPC (posterior) eigenvalue plot based on a priori model. (**D**) Loading plot of most impacting variable (arbitrary threshold set to 0.02).

**Figure 2 nutrients-15-05026-f002:**
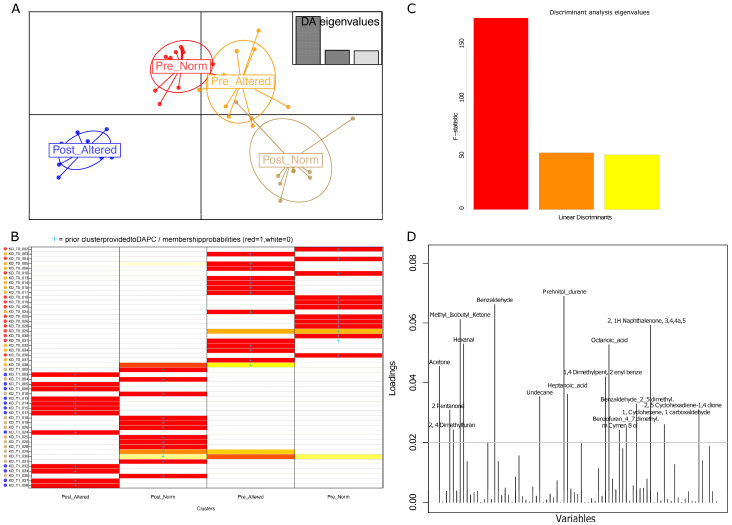
Urine metabolomics DAPC clustering. Discriminant analysis of principal components of volatile organic compounds in obese patients with altered and normal intestinal permeability before (pre) and after (post) VLCKD administration. (**A**) DAPC plot based on discriminant eigenvalues. (**B**) proportions of successful reassignments (based on the discriminant functions) of individuals to their original cluster. Red cells indicate a single group match derived from the “a posterior” reassignment. (**C**) DAPC linear discriminants. (**D**) Loading plot reporting the contribution of most impacting variables (VOCs) useful in discriminating sample clusters.

**Figure 3 nutrients-15-05026-f003:**
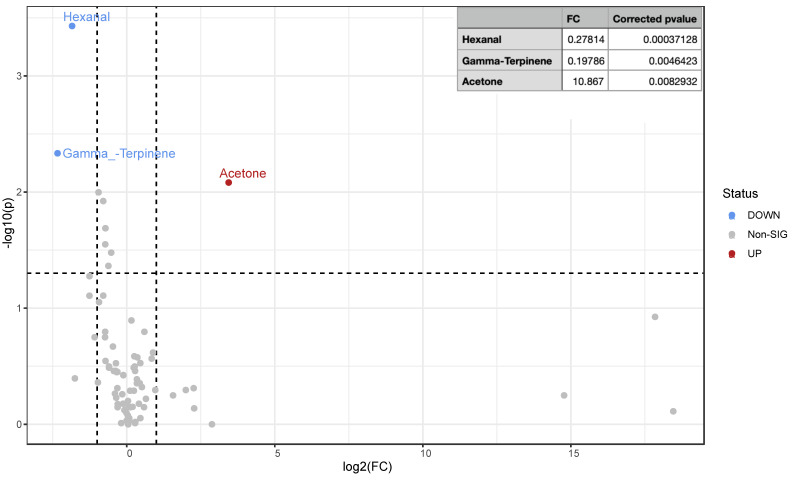
Volcano plot of statistically significant urinary VOC fold changes in patients marked by altered intestinal permeability (post-altered versus pre-altered). VOCs in urine samples from obese patients marked by altered intestinal permeability were compared before and after VLCKD administration via a non-parametric Wilcoxon rank-sum test combined with a fold change analysis. Due to the group comparison direction, increased (up, red) and decreased (down, blue) VOC concentrations are relative to altered phenotype after treatment. Under threshold, VOCs are reported as grey dots. The table below the volcano plot reports significant VOC fold change as raw values (FC), log2 fold change (log2FC), corrected *p*-values, and the *p*-value base-10 logarithm.

**Figure 4 nutrients-15-05026-f004:**
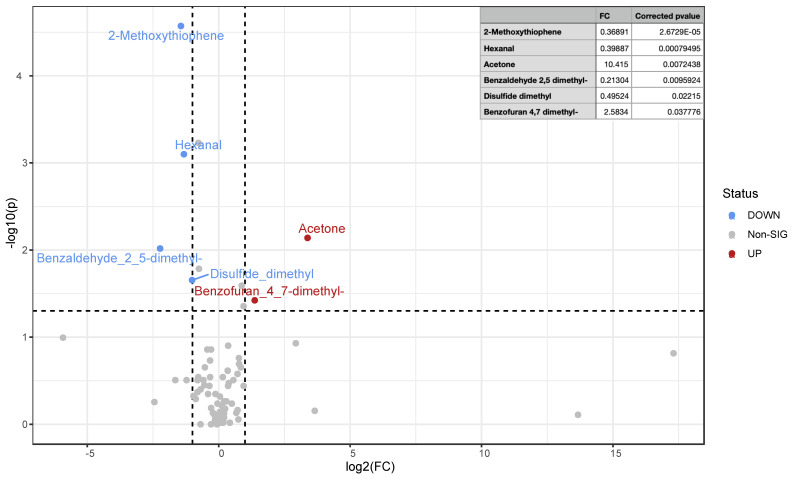
Volcano plot of statistically significant urinary VOC fold changes in patients marked by normal intestinal permeability (post-normal versus pre-normal). VOC in urine samples from obese patients marked by normal intestinal permeability was compared before and after VLCKD administration via a non-parametric Wilcoxon rank-sum test combined with a fold change analysis. Due to the group comparison direction, increased (up-red) and decreased (down-blue) VOC concentrations are relative to altered phenotype after treatment. Under threshold, VOCs are reported as grey dots. The table below the volcano plot reports significant VOC fold change as raw values (FC), log2 fold change (log2FC), corrected *p*-values, and the *p*-value base-10 logarithm.

**Figure 5 nutrients-15-05026-f005:**
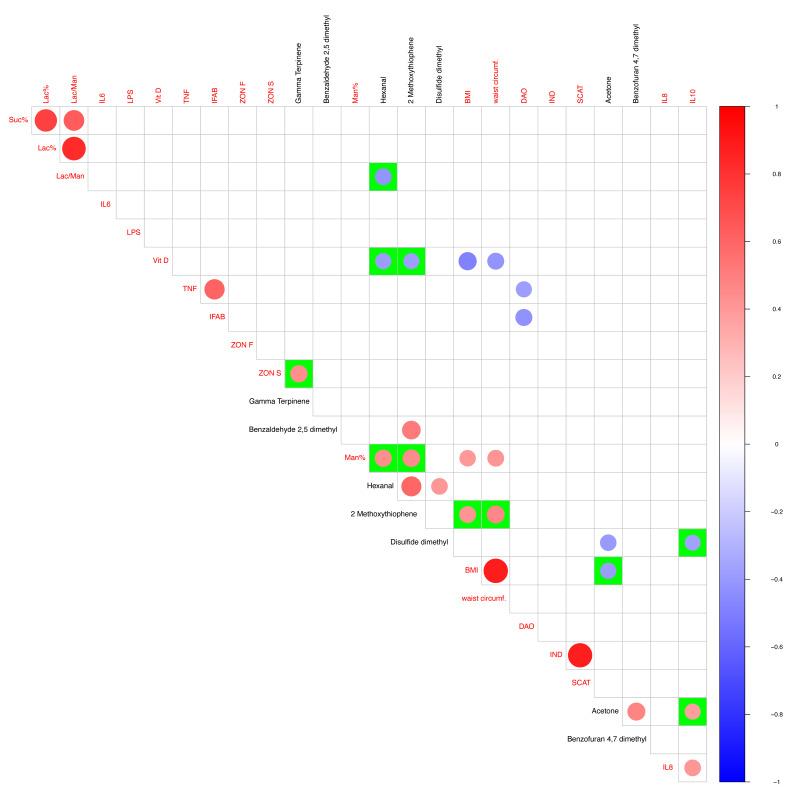
VOC and clinical/anthropometric parameter correlation. Pearson’s correlation between statistically significant VOCs (non-parametric Wilcoxon rank-sum test/fold change) from urine samples and clinical/anthropometric gathered parameters. Only statistically significant correlations (*p*-value < 0.01) have been reported. Pairwise compared variables from clinical/anthropometric (red font) and GC-MS (black font) have been plotted on green background squares. Red to blue scale bar colors are meaningful of positive and negative correlations. The size of the circles is proportional to the correlation value, both in case of positive and negative correlations.

## Data Availability

The datasets used and/or analyzed during the current study are available from the corresponding author upon reasonable request. The data are not publicly available due to privacy.

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
