# Peer review of "Metabolomic Profiling of Obese Patients with Altered Intestinal Permeability Undergoing a Very Low-Calorie Ketogenic Diet"

_nutrients, 2023, doi:10.3390/nu15245026_

Round 1

Reviewer 1 Report

Comments and Suggestions for Authors

In the manuscript “Metabolomic Profiling of Obese Patients with Altered Intestinal Permeability Undergoing a Very Low Carbohydrate Ketogenic Diet (VLCKD)”, Francesco Maria Calabrese, et al. collected urinary and fecal samples from obese patients with both normal and altered permeability, investigate the impact of altered permeability in obese patients undergoing a calorie-restrictive dietary regimen (VLCKD). In general, authors explored the impact of an 8-week VLCKD on obese patients with altered and normal intestinal permeability However, the article shows more formatting and content layout errors, and appropriately revisions are suggested.

Questions were raised as below and needed to be addressed.

1.       Figure 1 needs to be revised, the resolution of the image is too low, the font colors of figures A, B, C, and D are too light and not uniform, and the legend is not clearly visible. The metabolite names on the left side of the Y-axis of Figure B should be aligned. Figures C and D lack legends.

2.       Figure 2 needs to improve the clarity. Figure B needs a legend added to label the grouping of the bars

3.       In Figures 3 and 4, the numbers on the Y-axis of the volcano plot overlap with the ticks. The presentation of the table is too monotonous and should be replaced with other diagrams, or other styles of tables.

4.       In Figure 5, the correlation diagram is inconsistent between the substances in the upper and diagonal edges, and the diagonal edge is missing "IL10"? What does the size of the ball in the figure represent? Please indicate in the legend.

5.       Do patients have a phenotype and target for measuring intestinal permeability and how to confirm whether permeability is altered or not.

6.       In this study, intestinal permeability was altered in obese patients receiving VLCKD, and some metabolite alterations were identified by metabolomic analysis. Still, more in-depth studies should be done, such as 16S or ITS sequencing to identify changes in gut microbes or RNAseq to determine the expression of certain characterized genes.

Author Response

Reviewer 1

In the manuscript “Metabolomic Profiling of Obese Patients with Altered Intestinal Permeability Undergoing a Very Low Carbohydrate Ketogenic Diet (VLCKD)”, Francesco Maria Calabrese, et al. collected urinary and fecal samples from obese patients with both normal and altered permeability, investigate the impact of altered permeability in obese patients undergoing a calorie-restrictive dietary regimen (VLCKD). In general, authors explored the impact of an 8-week VLCKD on obese patients with altered and normal intestinal permeability However, the article shows more formatting and content layout errors, and appropriate revisions are suggested. Questions were raised as below and needed to be addressed.

Query 1. Figure 1 needs to be revised, the resolution of the image is too low, the font colors of figures A, B, C, and D are too light and not uniform, and the legend is not clearly visible. The metabolite names on the left side of the Y-axis of Figure B should be aligned. Figures C and D lack legends.

Reply 1. We thank the reviewer for pointing out this issue. Dealing with resolution, we directly uploaded the pdf version of the figure 1 in the answering form. Some problems with pixel and resizing of the figure may be occurred due to the word format used in the submission. We adjusted the legends relative to parts C and D, and we increase the contrast of colors in the DAPC plot.

Query 2. Figure 2 needs to improve the clarity. Figure B needs a legend added to label the grouping of the bars.

Reply 2. We thank the reviewer for highlighting this lack. We improved the legend relative to panel B and added colored point in order to easily connect samples to their own a priori assigned group.

Query 3. In Figures 3 and 4, the numbers on the Y-axis of the volcano plot overlap with the ticks. The presentation of the table is too monotonous and should be replaced with other diagrams, or other styles of tables.

Reply 3. We thank the reviewer for this observation. Following her/his comment we modified figures 3 and 4 by simplifying the two tables (only p value and FC columns were maintained) and embedding them in the volcano plot.

Query 4. In Figure 5, the correlation diagram is inconsistent between the substances in the upper and diagonal edges, and the diagonal edge is missing "IL10"? What does the size of the ball in the figure represent? Please indicate in the legend.

Reply 4. We are sorry for not being clear in the figure legend. The circle size is indicative of the value of correlation. In other words the size is directly proportional to the correlation value. The diagonal (self-correlations) has not been reported in the correlation matrix. IL10 variable is present only on the x axis of the correlation matrix and in its column all the variables (as rows) have been compared with.

Query 5. Do patients have a phenotype and target for measuring intestinal permeability and how to confirm whether permeability is altered or not?

Reply 5. We thank the reviewer for have risen this issue. In individuals with obesity, several studies have elucidated modifications in gastrointestinal functions, encompassing changes in the integrity and functionality of the intestinal barrier. The Lactulose/Mannitol ratio stands out as the preeminent test employed in human subjects to ascertain the permeability of the small intestine. Utilizing two distinct markers, a monosaccharide and a disaccharide, eliminates confounding factors associated with absorption, distribution, or excretion. Patients exhibiting a Lactulose/Mannitol ratio equal to or exceeding 0.030 are designated as experiencing alterations in permeability (please see lines 132-134 and 196-198).

Query 6. In this study, intestinal permeability was altered in obese patients receiving VLCKD, and some metabolite alterations were identified by metabolomic analysis. Still, more in-depth studies should be done, such as 16S or ITS sequencing, to identify changes in gut microbes or RNAseq to determine the expression of certain characterized genes.

Reply 6. Many thanks for this comment. As suggested, this and other study limits have been summarized in one extra paragraph reporting the points of weakness of the paper. We are planning to extend our cohort and perform a 16S sequencing experiment.

Reviewer 2 Report

Comments and Suggestions for Authors

Dear Authors,

thank you for this interesting manuscript written in good and understandable English.

Please explain the space you gave microbiome/microbiota in the introduction without measuring it. Maybe you could insert more information about VOCs fitting your results and your discussion.

Could you please add the male/female ratio to the groups? Please give also the average age of the groups for better comparison.

Line 180: Could you please specify "ketogenesis was present in almost all the patients"? How did you handle the data of the patients where ketogenesis was not present?

Please explain why you did not adjust the sample of 20% (w/v) chlorohexidine to the given urine volume. This could influence the results of the urine examination. Why did you put the antibiotics in the urine? What do you want to prevent?

Line 187: Please correct to chlorhexidine

Statistics: Did you pair the result of one person before and after the diet? I would suggest to analyse the changings in every patient belonging to the group to clarify the effects of the diet and the biological relevance of the changings. If this relevance cannot be proven the markers cannot help. Could you please clarify in the text?

I am sorry I can hardly see the Figure1 and Figure 2 B,C,D. The lines look very blurred and so I cannot read anything here. Could you please check? Maybe this is a problem of my computer.

Could you please set a ± in between the numbers in the tables? This would make the table more readable in my opinion. (To get the symbol hold ALT and press 0177.)

Line 451 Please explain how to bring the VOCs together with microbiota.

Line 468 Please include the influence of age, gender and lifestyle of the patients

Line 478 Please set i)

Line 523 Please set E.coli to italic letters

Line 530 Please set Curcuma longa to italic letters

Please summarize the study limitations in one extra paragraph

Reference 47: Please check the names. There are only abbreviations.

Could you think of measuring the VOCs in the breath in further studies? In my opinion this might be a easy and fast method and is easier to equalize between all patients sample collection as taking urinary samples by the patients themselves might be conducted unequal and therefore influence the results significantly.

Author Response

Reviewer 2

Dear Authors,

thank you for this interesting manuscript written in good and understandable English.

Query 1. Please explain the space you gave microbiome/microbiota in the introduction without measuring it. Maybe you could insert more information about VOCs fitting your results and discussion.

Reply 1. We thank the reviewer for her/his useful correction. As suggested, we gave less emphasis to microbiota by trimming and modifying the last sentence of the introduction section. Please see lines 120-123

Query 2. Could you please add the male/female ratio to the groups? Please also give the average age of the groups for better comparison.

Reply 2. As requested by the referee, the information about sex and age has been added in the new version of the manuscript body text. Please see lines 283-285 in the result section.

Query 3. Line 180: Could you please specify "ketogenesis was present in almost all the patients"? How did you handle the data of the patients where ketogenesis was not present?

Reply 3. Thank you. Actually, the sentence is incorrect because the adverb “almost” was a typo that did not correspond to reality, as ketogenesis was obtained in all patients. For this reason, the sentence was rewritten as follows: “Ketosis status was determined by monitoring markers in urine samples collected during the diet. Because we used a semi-quantitative method, this finding confirms that ketogenesis was present in all the patients and allowed us to define their diet as ketogenic" (lines 183-186).

Query 4. Please explain why you did not adjust the sample of 20% (w/v) chlorohexidine to the given urine volume. This could influence the results of the urine examination. Why did you put the antibiotics in the urine? What do you want to prevent?

Reply 4. Thank you. Chlorohexidine is a synthetic compound exhibiting disinfectant properties with a wide-ranging efficacy against Gram-positive and Gram-negative bacteria. Furthermore, it demonstrates effectiveness against select viruses and fungi. Its application was employed to mitigate microbial contamination risk that could compromise test results. Through our empirical observations, considering an average urine collection volume of approximately 400-500 ml, we have determined that 1 ml of chlorhexidine proves adequate in preventing contamination, irrespective of the final urine volume.

Query 5. Line 187: Please correct to chlorhexidine

Reply 5. As requested, the correction has been made.

Query 6. Statistics: Did you pair the results of one person before and after the diet? I would suggest to analyse the changings in every patient belonging to the group to clarify the effects of the diet and the biological relevance of the changings. If this relevance cannot be proven, the markers cannot help. Could you please clarify in the text?

Reply 6. Many thanks for have raised this point. We were interested in measuring the effect of the VLCKD on altered and normal permeability patients separately. The single paired results are in our analyses to much heterogeneous and in order to counteract fluctuation in variable distribution we usually run two- paired corrected group test. This allows for lowering the background noise and estimate the statistical error by using a higher number of samples. We thus carried out metabolomics on the same patients before and after the VLCKD and then we applied nonparametric corrected Wilcoxon rank-sum test (Benjamini-Hochberg) combined with a fold change analysis. Statistical test details are reported in M&M at line 266-268.

Query 7. I am sorry I can hardly see Figure 1 and Figure 2 B, C, D. The lines look very blurred, so I cannot read anything here. Could you please check? Maybe this is a problem of my computer.

Reply 7. Many thanks. We are sorry for this problem. The quality in the word file is not so high. We prepared a pdf file that we attached to the rebuttal form.

Query 8. Could you please set a ± in between the numbers in the tables? This would make the table more readable in my opinion. (To get the symbol hold ALT and press 0177.)

Reply 8. Many thanks. As requested, the symbol ± has been set in between the numbers in the tables

Query 9. Line 451 Please explain how to bring the VOCs together with microbiota.

Reply 9. Many thanks.

Briefly we added in the text the VOC/microbiota link explanation. Please see line 478-481. Moreover as requested by another reviewer, we dampened the microbiota part in the introduction section.

Query 10. Line 468 Please include the influence of age, gender and lifestyle of the patients

Reply 10. Many thanks we modify the sentence in the body text. Please see lines 525-526

Query 11. Line 478 Please set ,

Reply 11. Thanks we added the comma.

Query 12. Line 523 Please set E.coli to italic letters

Reply 12. As requested, “E. coli” has been re-written in italics.

Query 13. Line 530 Please set Curcuma longa to italic letters

Reply 13. As requested, “Curcuma longa” has been re-written in italics.

Query 14. Please summarize the study limitations in one extra paragraph

Reply 14. As requested, study limits have been summarized in one extra paragraph reporting the points of weakness of the paper. Based on the journal guidelines it is not possible to insert a “limitation” paragraph, so we put limitations at the end of the discussion section.

Query 15. Reference 45: Please check the names. There are only abbreviations.

Reply 15. Many thanks. Reference 45 has been checked for names and reported according to the Journal’s format.

Query 16. Could you think of measuring the VOCs in the breath in further studies? In my opinion this might be a easy and fast method and is easier to equalize between all patients sample collection as taking urinary samples by the patients themselves might be conducted unequal and therefore influence the results significantly.

Reply 16. Many thanks for this important consideration. We totally agree on this point. In our experimental design, we planned to measure the effect of diet by measuring fecal and urine metabolite GC/MS. Moreover, our future purpose is to link the fecal microbiota with these metabolites (fecal metabolomics). We are really interested in breath metabolomics and our department facility is moving towards the set of breath metabolomics, that requires different methods of extraction based on different columns. Thus we plan to carry out also this analysis related to a specific branch of metabolomics in the next future.

Round 2

Reviewer 1 Report

Comments and Suggestions for Authors

No more comments

Reviewer 2 Report

Comments and Suggestions for Authors

Dear Authors,

Thank you for taking my recommendations into account! In my opinion the manuskript is now ready for publication.

kind regards